# Panic in the Pandemic: Determinants of Vaccine Hesitancy and the Dilemma of Public Health Information Sharing during the COVID-19 Pandemic in Sri Lanka

**DOI:** 10.3390/ijerph21101268

**Published:** 2024-09-24

**Authors:** Thushara Kamalrathne, Jayasekara R. Jayasekara, Dilanthi Amaratunga, Richard Haigh, Lahiru Kodituwakku, Chintha Rupasinghe

**Affiliations:** 1Global Disaster Resilience Centre, School of Applied Sciences, University of Huddersfield, Queensgate, Huddersfield HD1 3DH, UK; dr.d.amaratunga@gmail.com (D.A.); r.haigh@hud.ac.uk (R.H.); 2Department of Civil and Environmental Engineering, Louisiana State University, Baton Rouge, LA 70803, USA; ravindujy@gmail.com; 3Ministry of Health, Colombo 01000, Sri Lanka; lahiru.kodituwakku@hud.ac.uk (L.K.); chintha.rupasinghe@hud.ac.uk (C.R.)

**Keywords:** vaccine hesitancy, vaccine hesitancy in Sri Lanka, determinants of vaccine hesitancy, COVID-19 pandemic, public health information, public health knowledge

## Abstract

Vaccine hesitancy, a pressing global challenge in vaccination programs, was significantly amplified during the COVID-19 pandemic. The proliferation of misinformation, including false claims and rumours, and the influence of anti-vaccine movements fuelled hesitancy. This study aims to explore the socio-economic determinants that influenced vaccine hesitancy and the impact of public health information sharing in Sri Lanka during the pandemic. The study employed a comprehensive mixed-method approach for data collection, administrating a household survey *(n* = 3330) and 206 semi-structured interviews. The survey results indicated that 37.8% (*n* = 3113) of respondents delayed or rejected vaccines for various reasons, the leading cause being the fear of side effects of the vaccine. Although fear of side effects was the prime reason for rejection (*n* = 1176, 46.29%), respondents demonstrated an extremely poor understanding of the potential side effects of vaccines, which was 55.39% (*n* = 3113). Notably, 84.60% (*n* = 3113) were unaware of the vaccine development process. Multivariate logistic regression analysis showed that middle-income people (AOR—0.42) and females (0.65) were less likely not to make decisions based on scientific information compared to underprivileged people and males. The survey also revealed that strong belief in the use of traditional medicines and remedies and religious beliefs (*n* = 1176, 3.95%) were among the main reasons for hesitancy. The findings illustrate that misinformation, lack of health knowledge, and lack of understanding to seek scientific information have fuelled vaccine hesitancy in Sri Lanka.

## 1. Introduction

The COVID-19 pandemic is the most devastating global public health catastrophe in recent history after the deadliest Spanish flu reported in 1918, in which approximately 50 million people were killed worldwide [1]. With the dramatic increase in SARS-CoV-2 infections and deaths, the World Health Organization (WHO) declared a public health emergency of international concern in January 2020 [2]. The pandemic response worldwide became intense as the SARS-CoV-2 virus continued to mutate and evolve into various distinct strains, some of which were highly contagious and evaded human antibodies [3]. Further, the gravity of the pandemic considerably intensified, and response and mitigation became challenging due to unmeasurable cascading impacts that extended broadly into macro environments such as social and economic sectors.

Vaccinations are believed to be one of the most outstanding achievements of public health, as vaccines have tremendously contributed to decreasing mortality and morbidity associated with thousands of infectious diseases throughout human history [4]. Further, vaccination against the SARS-CoV-2 virus has been identified as a critical medical countermeasure under the pandemic preparedness and response strategy to mitigate the risks related to COVID-19 deaths and infection [5]. COVID-19 vaccination has been executed outside the clinical trials since 8 December 2020 [6]. Undoubtedly, the vaccine rollout has saved millions of lives so far. Using mathematical modelling, a study emphasised that COVID-19 vaccination prevented approximately 14.4–19.8 million deaths in around 185 countries and territories during only the first year of the global vaccination programme [6]. According to the World Health Organization, 13.3 billion vaccine doses have been administrated worldwide against the SARS-CoV-2 virus [7]. Approximately 5.5 billion people have been vaccinated, while health authorities of partner countries of the WHO have fully vaccinated 5.1 billion people. In general, 70.1% of people have received at least one dose of COVID-19 vaccines, whereas this value is 30.2% in low-income countries [8]. Evidently, the death toll of the unvaccinated was higher than that of the vaccinated during the COVID-19 pandemic [9].

Vaccine hesitancy, which refers to a delay in acceptance or refusal of vaccines despite the availability of vaccination services [9], has been one of the critical challenges in administrating vaccination programmes globally. Misinformation is one of the primary causes of vaccine hesitancy, which might surface in different forms, including false claims, disinformation, and rumours [10,11,12,13]. Anti-vaccine activism, another challenge encountered during vaccination which causes hesitancy, was initiated in every corner of the world by diverse groups when vaccination programmes started on a large scale [14,15]. Many anti-vaccine campaigns were incepted based on the vaccine’s presumed side effects, vaccine manufacturing technologies [16], possible herd immunity, religious thoughts, anti-Western and anti-capitalistic ideologies, and some myths [17,18]. As a clearly visible trend, online-based communication platforms, including social media, played a critical role in disseminating such information which encouraged vaccine hesitancy [19,20,21,22] However, these anti-vaccine ideologies and campaigns lowered the efficiency of vaccination programmes everywhere by fuelling uncertainties among people regarding vaccination [19]. Reducing vaccination coverage and delaying the reach of herd immunity are impermeable sides of anti-vaccine activism [18]. In this milieu, this research provides deep insights into the COVID-19 vaccination programme conducted in Sri Lanka relevant to its public response, causes of vaccine hesitancy, vaccine preference, and the status of public health information sharing among communities. The objective of this study is to explore socio-economic determinants that affected vaccine hesitancy and public health information sharing during the COVID-19 pandemic in Sri Lanka.

### 1.1. Overview of Vaccine Hesitancy

Vaccines are considered an essential tool which can improve health outcomes and prevent death. However, this can only be achieved if countries sustain high vaccine coverage and vaccine uptake rates. Over the years, vaccination campaigns with high and sustained coverage have achieved many milestones, including the Polio elimination and Smallpox eradication programmes [23]. However, in recent years, these global accomplishments have been put to the test due to increasing ‘Vaccine Hesitancy’ among numerous ethnic, religious, and socio-economic groups across the globe [24]. This phenomenon was evident during the COVID-19 pandemic when many governments had to strategize their vaccination campaigns to meet the new challenge of vaccine hesitancy [25].

Globally, one in five children does not receive routine and lifesaving vaccinations, whereas an estimated 1.5 million children die yearly from vaccine-preventable diseases [26]. This has a direct and lasting impact on disease elimination and eradication programmes around the globe, adding an extra burden on health systems for treatments and hospitalisations due to vaccine-preventable diseases.

However, this phenomenon is not straightforward, as the definition of vaccine hesitancy by the WHO’s Strategic Advisory Group of Experts on Immunization (SAGE) suggests: “*A delay in acceptance or refusal of vaccines despite availability of vaccination services. Vaccine hesitancy is complex and context-specific, varying across time, place, and vaccines. It is influenced by factors such as complacency, convenience, and confidence*” [24]. It is enmeshed in a complex web of reasons traversing the entire spectrum of social determinants of health [27]. Vaccine hesitancy is not only confined to socially disadvantaged and economically weaker population cohorts with limited access to affordable health care but is also prevalent among the people in higher tiers of social and economic classes with higher levels of health education. Further, researchers have argued that the phenomenon cuts across the individual, community, and country levels [28].

However, the World Health Organization identifies three primary constructs (3C model), Confidence, Convenience, and Complacency, for vaccine hesitancy among many diverse contextual reasons in different countries [29]. Considering a large amount of empirical evidence, the 3C model was revised in 2018 as the 5C model, adding two significant elements: Calculation and Collective responsibility [24].

Confidence—Lack of confidence in the vaccine’s effectiveness, efficacy, and safety, as well as the competency of health care workers (HCWs) who deliver the vaccines. The motivation and objectives behind the vaccination campaigns are often questioned.Convenience—Questions on being physically able to visit the site of vaccine delivery, affordability of the vaccine, understanding the instructions given (Health Literacy), and acceptance of vaccination within the cultural norms.Complacency—Ability to perceive the risk of getting infected with the disease vs. being protected by the vaccine.Collective Responsibility—Willingness to protect others.Calculations—Engagement in gathering extensive information.

The Vaccine Hesitancy Determinants Matrix [30], another model developed to understand the phenomenon, describes all these probable causes in three major categories, namely:Contextual—Communication and media environment, influential leaders/gatekeepers, religion/culture/gender/economy/social status, politics, and policies.Individual/Group—Beliefs and attitudes on prevention, personal or family experiences in previous vaccinations, perceived risk and benefit, trust in the health care system and HCWs, etc.Vaccine/Vaccine Specific Influencers—Vaccine schedule, cost, design, and mode of delivery in the vaccination programme, reliability of vaccination equipment and the strength of recommendation by a health care professional.

Vaccine hesitancy is driven by various physical, psychological, socio-demographic, and other cultural determinants. A previous study identified 72 such barriers, under physical, contextual, psychological, and socio-demographic determinants, that impact the hesitancy for the influenza vaccine [31]. Further, studies have emphasised the significance of the psychological aspect, which is determined by utility, past behaviours, experience, and knowledge, in understanding vaccine acceptance [23].

Generally, three categories of unvaccinated individuals have been identified by national and global health authorities: unvaccinated and not willing to get vaccinated, unvaccinated and uncertain if willing to get vaccinated, and unvaccinated and willing to get vaccinated [9]. Research suggested that the proportion of people who are unvaccinated and not willing to get vaccinated in Germany is 19.26%, whereas in the Netherlands it is 17.76%; UK, 17.10%; Japan, 12.77%; and Spain, 9.22% [8]. Lazarus et al [32] found that vaccine hesitancy is a growing concern among eight countries, where they included 23 in their study. Accordingly, the UK (1.0%), China (1%), Turkey (2.7%), Brazil (3.3%), Kenya (8.5%), Mexico (9.4%), Ghana (13.8%), and South Africa (21.1%) were the countries in which hesitancy has soared extraordinarily.

Vaccine intake in countries differs for several economic, social, and political reasons [33,34]. For example, Africa has the lowest vaccinated number (30.58), whereas the Western Pacific has the highest vaccinated number among WHO regions regarding persons vaccinated with a complete primary series per 100 population. Also, these two regions are the lowest and highest in total doses administrated per 100 population, 52.75 and 241.22, respectively. South-East Asia reports 164.7 total doses (per 100 population), which is lower than the global average of 171.87 [7]. Another piece of research suggested that Africa and Asia have the least vaccine acceptance in terms of regions [34]. Among its neighbouring nations, Sri Lanka is the third lowest in terms of persons vaccinated with the primary complete series of vaccines during COVID-19 [35].

### 1.2. COVID-19 Pandemic and Vaccination Status in Sri Lanka

Sri Lanka’s first infected case was reported on 27 January 2020 after an index patient who temporarily migrated from China was diagnosed as positive for COVID-19. Thereafter, the first local infected case was reported on 11 March 2020, who was a tourist guide working with an Italian tourist group [36,37]. Gradually, the virus spread, beginning with disease clusters across the country. Compared with neighbouring countries, managing virus transmission and deaths during the first and second waves of the COVID-19 pandemic was very successful and remarkable, given the resource-constrained environment [38,39].

As of 4 October 2020, only 3346 infected cases and 13 deaths were reported during the first wave. However, a gradual relaxation of strict quarantine and mobility restrictions led to a notable increase in patient numbers during the pandemic’s second wave, which resulted in approximately 89,626 positive cases [35,40]. The impact of the third wave was devastating for the population and the health system, with 672,215 positive cases and 16,847 deaths reported as a total due to the COVID-19 pandemic by 8 May 2023 [35]. Colombo, Gampaha, and Kalutara were the severely affected districts in Sri Lanka during all three waves of the pandemic.

National data suggest that the deaths of males (9500), which is 56.3%, are higher than those of females (7347). More COVID-19 deaths have been reported from those over 60 or immunocompromised individuals suffering from underlying health conditions. Figure 1 shows the distribution of COVID-19 deaths among the age cohorts.

According to the Sri Lankan epidemiological data, 78.21% have taken the first dose, whereas 67.32% have accepted the second dose. However, the first booster has been accepted only by 46.71%, while the second booster has been taken only by 1.18% (Table 1) [35]. A few studies conducted in Sri Lanka on vaccine acceptance administrating a considerably large sample of respondents suggest that vaccine hesitancy among the population is significant [41,42].

## 2. Materials and Methods

A mixed-method approach was used to collect primary data in this research to investigate the social factors that affected vaccine hesitancy during the COVID-19 pandemic and explore the challenges of public health information sharing among underprivileged groups in Sri Lanka during the pandemic. A household survey was carried out to collect quantitative data, and semi-structured interviews were conducted to collect qualitative data. Fifteen trained research assistants (RAs) were involved in data collection, carried out over three months, from 5 July 2022 to 16 October 2022. Research assistants were individually involved in data collection in distinct locations.

### 2.1. Household Survey

Quantitative data were collected from 3330 households in 26 selected divisional secretariats (DSs) of 9 districts in Sri Lanka. The total population of Sri Lanka was 22.181 million for the year 2022 [43], in which the field study was started. Cochran’s sample size formula for categorical data was used to calculate the minimum sample size. The formula yields a minimum sample size of 384 for any population over 200,000 with a marginal error of 5% and a confidence interval of 95% [44]. DSs were selected based on the socio-cultural and economic diversity among the population and physical remoteness in selected areas. The field study covered three distinct sectors of residents, rural, urban, and estate, to reflect the variation in determinants among sectors and weigh the population of the respective resident sectors. Table 2 presents the selected divisional secretariats in respective districts. The inclusion criteria used to select respondents for data collection were being more than 18 years old and consent to be part of the study. The study did not apply exclusion criteria, except the participant’s age, to explore the diverse groups and their responses towards vaccination.

The multi-stage stratified cluster method was used as the sampling technique. Main clusters were developed based on rural, urban, and estate residence types (Appendix A). Households were randomly selected using a household list, and only one respondent over 18 was selected from each household to collect data for the survey, considering the inclusion of gender and age according to the sample calculation. After that, sub-clusters were further identified based on ethnicity, gender, and age. Two DSs were purposively selected to include Indigenous communities in the sample. However, the Indigenous community was not considered a distinctive sector as many of these communities live in remote rural areas. A middle-income and low-income cluster was identified within a one-kilometre radius within selected areas to understand the economic variation and determinants.

A digital questionnaire was developed using the SurveyMonkey application (SurveyMonkey Inc., San Mateo, CA, USA (Available online: www.surveymonkey.com, 10 January 2022) The Questionnaire consisted of 50 questions focused on six sections: (1) demographic data of respondents, (2) prevailing health conditions and regular medication, e.g., blood sugar and cholesterol, cardiovascular, kidney disease, etc., and (3) vaccine data, i.e., number of doses, (4) respondents’ knowledge about vaccines, such as types, manufacturing and technology, and effectiveness, (5) their knowledge about adverse impacts, and (6) opinion about public health information sharing during the COVID-19 pandemic (Appendix A). Each RA physically visited households to record responses using a digital device to input participants’ responses to the database administered by SurveyMonkey.

The investigators developed the questions as there were no validated tools for assessing vaccine hesitancy in Sri Lanka. The questions were formulated to meet the specific objectives of the study. A comprehensive literature survey was performed, and the technical expertise of the field was used for constructing a validity assessment. In addition, the technical experts involved in the COVID-19 response from relevant departments [e.g., the Epidemiology unit of the Ministry of Health, etc.] were consulted in assessing the face and content validity of the questionnaire. The questionnaire was compared with the available literature and survey questionnaires to assess the concurrent and content validity. The questionnaire was formulated in English and translated into the Sinhala and Tamil languages. Simple language was used in translations to maintain the feasibility of self-administration. The completed questionnaires were back-translated to English by different translators to ensure the content of the tool remained intact.

The survey was pretested among fifty volunteers of the Kandy district and was updated following the pretest. The pretesting aimed to determine the questionnaire’s feasibility and acceptability. Focused small group discussions were held with the pretest participants and they were inquired about the understandability of the questions and wordings and the time needed to answer the questionnaire. Following this, the ambiguous questions and wordings were identified and modified accordingly. The pretested population was similar to the study population and represented urban, rural, and estate populations. All three languages were pretested and forwarded for ethical review. The pretested questionnaire was reviewed by the ethical committee and revised according to their guidance. A revised version of the questionnaire was again pretested and updated among 30 volunteers in the Kandy district. The revised version of the questionnaire was used in the data collection process.

The researchers closely monitored the database during the data collection period to minimise technical errors. Once the survey was completed, the data set was rechecked for any errors before proceeding to the analysis. Another team member reviewed the updated data set again to minimise errors before analysis.

### 2.2. Statistical Analysis

A total of 3330 responses were received, and the response rate was 100%. However, 217 responses were excluded due to incomplete responses, so the final sample consisted of 3113 responses. The Statistical Package for the Social Sciences [SPSS] v.25 and Microsoft Excel (v.2406) were used to analyse quantitative data. For the selected variables, inferential statistics were used to elicit associations.

All the variables were considered as categorical variables in the statistical analysis. Initially, the Chi-square test was used to investigate possible differences between groups of variables. Based on the results, univariate and multivariate logistic regression analyses were carried out to elicit associations with the dependent variables, such as knowledge of COVID-19 vaccines and the use of scientific information for decision-making. The participants’ genders, income levels, living areas, ages, and religions were used as independent variables in the logistic regression. The significance of the associations was sought statistically using the regression analysis, and the confidence level was taken as 95% [α = 0.05]. A significance level [*p* value] of <0.05 was considered as having a significant association. The level of significance was taken as 0.05. The results of the logistic regression analyses were expressed as crude odd ratios (CORs) or adjusted odd ratios (AORs) with 95% confidence intervals (95% CIs). McFadden’s Pseudo R-squared value [45] was used as the goodness-of-fit measure for the logistic regression models. Analysed data were summarised into tables and graphs using MS Excel and formatted to achieve the required format for manuscript submission.

### 2.3. Semi-Structured Interviews

Representing categories of clusters mentioned above, 208 semi-structured interviews, eight from each DS, were conducted simultaneously with the survey. Respondents were selected purposively, reflecting diversity of age, gender, and ethnicity. Each interview spanned between 45 and 60 min, and interviews were recorded on a mobile phone recorder after obtaining consent from the interviewee. An interview guideline was constructed with a major focus on (1) opinions about the side effects of the COVID-19 vaccines, (2) the level of satisfaction with health information received from authorities during the pandemic, and (3) the impact of misinformation and disinformation on vaccination.

The recordings were initially analysed using qualitative coding, followed by thematic analysis. The initially identified themes were reviewed by re-reading and recoding. The identified themes were defined and named accordingly before proceeding to the presentation of the results.

### 2.4. Ethical Considerations

The research was conducted as part of a project on Improving COVID-19 and pandemic preparedness and response through the downstream of multi-hazard early warning systems, led by the University of Huddersfield, UK. Ethical clearance was given to the project from the University of Huddersfield, and further clearance for field data collection was granted by the Ministry of Health, Sri Lanka. The National Dengue Control Unit, Ministry of Health, Sri Lanka, guided and monitored the entire process of the fieldwork.

Prior to recruitment, all the participants were given a brief introduction to the survey, including the objectives and study protocol. They were also provided with a declaration of anonymity and privacy. The survey was conducted in accordance with the Helsinki Declaration, and participants were not compensated in any way for their participation.

Written informed consent was obtained from survey respondents and interviewees before commencing data collection. Research Assistants (RAs) clearly explained the purpose of the research and ethical considerations of the study to respondents, especially those who had difficulty reading. Also, the support of two interpreters was obtained during data collection in the Indigenous settings to comply more with their language. As the Ministry of Health, the apex body regulating public health in Sri Lanka, issued clearance to the research, specific approval was not required to conduct field studies in Indigenous settings.

## 3. Results

### 3.1. Factors Affecting Delayed Vaccination and Rejection of Vaccination by the Respondents

Overall, as a percentage (%) of the sample (*n* = 3113), 7.53 have accepted only the first dose, and 55.25 have taken up to the second dose. The third dose, or the first booster, has only been accepted by 32.47, whereas 4.75 respondents have not taken any vaccines (Table 3).

As a clear pattern, vaccine rejection and delay significantly soared after completing the first and second vaccines, known as the primary complete series, as presented in Table 3. Interview data suggested that people accepted the primary complete series of vaccines due to extreme fear of virus infection and death. Further, several community interviews revealed that some people had accepted the primary complete series of vaccines because they were frightened of being in mass quarantine centres established hundreds of miles away from their villages. The following section elaborates on the reasons admitted by respondents to delay or reject vaccines. Interview data were fed into each section to deliver a descriptive aspect of the analytical factor.

The survey inquired about the reasons that people admitted to delaying or rejecting the vaccine once health authorities incepted the mass vaccination programmes in the country. The most outstanding reason for vaccine hesitancy was the fear of the vaccine’s side effects (*n* = 1176, 46.29%) (Table 4), known as adverse impacts or effects, imparted through a third party, i.e., social media or peers.

Interestingly, most of the respondents who mentioned that fear of the side effects was the key reason not to take or delay the vaccine mentioned that the dysfunction of the organs was the most worrying presumed side effect among several others, such as being infertile, cardiovascular diseases, and fatigue. Qualitative data suggest that the key concern among the younger group in terms of presumed side effects is fertility-related or dysfunction of reproductive organs. This combination is prevalent among unmarried and newly married couples expecting a child.

“*I am newly married, and my husband and I are expecting a baby. We have been planning a baby for the past three years. We met several doctors and did some religious and ritualistic performances to have a child without further delay. Several of our colleagues and relatives advised us to consider this situation before getting a vaccine because many people suspect that vaccine impacts people’s fertility. This may be false or a rumour, but we have no option. No one knows what is happening…*”(Female, age 27, urban, higher education, a public servant)

In contrast, those aged 50 and above had more concerns about fatigue, heart attacks, strokes, allergies, and non-fertility-related effects that were assumed to be some of the vaccine’s side effects, informed by rumours and other informal channels.

“*I am suffering from high blood pressure, cholesterol and blood sugar. Also, I have gone through a stem treatment for blocking a valve in the heart. Several educated people, including some professionals, said that some vaccines make blood clots, which will harm patients with cardiovascular treatments. So, I went completely insane because of this information and even had a long time to get the first vaccine. After seeing people getting vaccines fearlessly, I decided to take the first two doses and did not get the third vaccine due to various concerns over different channels…*”(Male, age 66, urban, higher education, a retired executive officer of public service)

The second most cited reason that respondents admitted was waiting to see the results of the vaccines (*n* = 1176, 17.40%). People who admitted this reason said they had a trust issue in the effectiveness of the vaccine against the SARS-CoV-2 virus and thus needed to see whether vaccines could decrease deaths. Interview data suggested that younger age cohorts (18–29) had this reason compared to other age groups.

“*I was waiting to accept the second dose because many information sources informed that there were discrepancies of vaccine’s effectiveness as the vaccine has produced without enough trials…*”(Male, age 25, higher education, urban)

Dissemination of negative information about the vaccines (*n* = 1176, 12.46%) was also a significant factor among respondents to delay or reject vaccines. Most of the respondents who admitted this reason highlighted that context of the negative information was based on conspiracy theory, i.e. vaccine manufacturers have hidden agenda on financial outcome and their future vaccine propaganda. Further, 4.27% (*n* = 1176) of respondents admitted that they did not have sufficient knowledge of the vaccines and thus delayed or rejected the vaccine.

Overall, 3.95% (*n* = 1176) of respondents answered that they delayed or rejected vaccines due to solid faith in traditional medicine or religious teaching. Interestingly, interview data suggest that the two communities have primarily been influenced by believing in traditional medicine and religious faith to not have vaccine doses. Some groups of an Indigenous community known as the Vedda community (Sri Lankan aboriginals) thoroughly believe in traditional medicine and rituals as the best cure for infectious diseases, which has been practised over a long period. They practice them when the spread of an infectious disease is noted in their setting.

Moreover, some religious communities have faith in religion in several areas, which is identified as a highly significant factor for vaccine hesitancy. For example, qualitative data suggested that Muslim communities have refrained from accepting vaccines in many areas believing vaccinees contain biological substances that are restricted to consume according to their religious faith. Also, there was a belief that vaccines decrease fertility, especially among females. Meanwhile. the government imposed the regulation of compulsory cremation for deaths caused by COVID-19 in Sri Lanka to minimise the spread of the disease. In this milieu, vaccine acceptance was sored among Muslim communities due to the fact that cremation is a spiritual barrier to achieve their destiny of life according to religious doctrine [46].

“*In the first stage, our community members hated to accept the vaccine because they thought it was against some religious principles. However, once cremation was started by the authorities, no one wanted to die as cremation was an extreme barrier to seeing the god. Thus, vaccine acceptance increased significantly…*”(Religious leader, age 64, rural)

### 3.2. Knowledge of Vaccines and the Vaccination Process

Knowledge about vaccines and the vaccination process, including the purpose of the vaccination and effectiveness against the SARS-CoV-2 virus, is essential in understanding the socio-economic factors affecting vaccine hesitancy in a broader spectrum. Therefore, the level of knowledge about vaccine types, vaccine production, and vaccine effectiveness were explored quantitatively and qualitatively (Appendix A). In general, knowledge about vaccines and vaccination among the sample population is significantly low in all aspects. This section employs the results of the multivariate logistic regression models for knowledge of vaccine effectiveness, side effects, suitability, and decision-making based on scientific information. The individual logistic regression models considered age, gender, living area, income level, and religion as the explanatory variables. The McFadden’s Pseudo R-squared values [45] for the fitted models were 0.12, 0.12, 0.13, and 0.11 for knowledge of vaccine effectiveness, side effects, suitability, and decision-making based on scientific information, respectively. The relatively low goodness-of-fit measures indicate that there might be other explanatory variables that affect the knowledge of vaccines among the community or that randomness in response variables is high. Therefore, logistic regression models were not used to predict the levels of knowledge of vaccines. However, these results were used to identify how the levels of individual explanatory variables affected the interested response variables.

#### 3.2.1. Knowledge of Vaccine Development, Country of Origin, and Manufacturer Process

It is believed that substantial knowledge of how vaccines are produced and what technology is used in vaccine manufacturing could enhance confidence among the public. In the survey, researchers tested the knowledge of vaccines among the sample based on respondents’ technical understanding of vaccines and whether they were aware of the vaccines from any other party. Accordingly, Sinopharm was the most well-known vaccine, followed by Pfizer and Moderna. From the total sample, 88.22% knew Sinopharm and 73.53% were aware of the Pfizer vaccine, whereas 50.64% of the sample knew Moderna. In contrast, Sinopharm and Pfizer are the most administrated vaccines in Sri Lanka (see Table 1). Interestingly, AstraZeneca was the third most administrated vaccine in Sri Lanka, but only 31.36% of respondents knew about the vaccine. Surprisingly, 6.13% of respondents from the total sample were unaware of any vaccine types used against the COVID-19 virus.

Participants’ knowledge of vaccine manufacturing processes and technologies was also explored. Overall, from the total sample (*n* = 3113), 84.60% responded that they were unaware of this element, 11.02% had little knowledge, and 3.09% were considerably knowledgeable. Only 0.10% of respondents admitted that they were fully aware of vaccine production and its technical aspects. Further, 35.89% of respondents did not know vaccine manufacturers or particular countries involved in vaccine production.

#### 3.2.2. Knowledge of the Vaccine Outcomes

Several questions were included in the survey to understand respondents’ knowledge of the function of vaccines against COVID-19. In total, 35% (*n* = 3113) of respondents replied that the purpose of the COVID-19 vaccines is to minimise complications, whereas 30.17% thought the purpose of the vaccine is to reduce the danger of death caused by the virus. The rest of the respondents believed that the vaccine could reduce the transmission of the disease, prevent virus infection, and prevent disease, as shown in Figure 2. Accordingly, a substantial number of respondents were unaware of what the vaccine is capable of.

In comparison, although the overall AOR trend showcased that females were more likely to have knowledge of vaccine suitability than males, a statistically significant high likelihood was observed among females with only a moderate knowledge level (AOR—1.81) (Table 5). When comparing knowledge quantiles among income groups, middle-income people had higher knowledge of vaccine effectiveness than underprivileged people (AORS—1.85, 2.33, 6.06, 16.37) (Table 6). In terms of the sector of residents, urban people showed more likelihood of having vaccine knowledge compared to village people (AORS—1.46 and 1.47).

It is further revealed that respondents’ knowledge of the effectiveness of vaccines against infection control and death was considerably low. There were no significant differences between genders in this variable. However, when comparing income groups, middle-income people were more likely to have higher knowledge of vaccine effectiveness compared to underprivileged groups (AORS—1.58, 2.12, 8.16, and 9.85) (Table 6). In terms of sector of residence, estate people had the poorest knowledge of vaccine suitability, and their knowledge level of vaccine effectiveness was also extremely low. In contrast, urban people maintained a consistent trend in having more likelihood to have low to moderate knowledge levels compared to village people (AORs 1.54 and 1.45) (Table 7).

Qualitative data suggest that respondents did not have adequate knowledge of the positive impacts of the vaccines, especially what protection can be expected for immunocompromised people and people with underlying health conditions, such as chronic diseases. This inadequate knowledge could primarily be seen among low-income groups in resource-limited settings such as urban low-income communities and estate communities. This knowledge gap between the underprivileged and middle-income communities can be seen in Figure 3.

As a pattern, when the respondent’s age increases, the level of knowledge of vaccine effectiveness gradually declines. Hence, it can be concluded that younger respondents’ knowledge of the effectiveness of the vaccines against COVID-19 is comparatively higher than that of the older respondents.

#### 3.2.3. Knowledge of Potential Side Effects of the Vaccine

Respondents were asked a set of questions to understand their knowledge of the potential side effects of vaccines. According to the survey results, the potential side effects of the vaccines have been identified as the key reason for vaccine hesitancy, as discussed in Section 3.1. Further, the same reason was highlighted by the respondents who had not taken a single dose of the vaccine. Accordingly, 58.97% (*n* = 148) of respondents who have not taken any vaccines emphasised that the fear of side effects was the key reason for them not to accept any vaccine shot, followed by advice received from third parties not to take vaccines. 

When comparing knowledge quantiles against gender, although females were less likely to have very low knowledge levels compared to males (AORS—0.74 and 0.84), no statistically significant trend was identified across other knowledge quantiles (Table 5). A large gap between residence sectors on the level of knowledge about possible side effects was observed, as shown in Table 7 Urban people were more likely to have knowledge levels from very low to moderate compared to village people (AORs—1.53, 1.38, and 1.92) (Table 7). Overall, the estate sector, where the majority is Indian Tamil, is the poorest in knowledge; almost 80% (*n* = 3113) of respondents were either entirely unaware or had too little knowledge about side effects. Also, this value is 65% and 60% in rural and urban sectors, respectively. More than 68% of respondents’ knowledge of possible side effects of vaccines is entirely inadequate.

The level of knowledge was further tested against the economic status of the respondents, which shows that low-income communities in rural, urban, and estate areas have unsatisfying knowledge levels in terms of vaccine’s potential side effects. Data indicate that middle-income people were likelier to have low to moderate knowledge levels compared to underprivileged people (AORs—1.33, 1.49, and 4.12) (Table 6). Remarkably, this inadequate knowledge of side effects can be seen among all age cohorts in the sample. Hence, there is no significant difference between age categories. Interview data suggested that females were more terrified of potential side effects than male interviewees.

“*I am 6 months pregnant. This is my first baby. I am scared because of side effects of vaccines that people talk about. I have a fear that if I take the vaccine, that will impact my baby…*”(Female, age 25, rural, secondary education, no occupation)

“*How can we believe the vaccine? There may be unseen side effects. Even paracetamol has side effects. Our peers were discussing the risk of malfunction or dysfunction of organs, especially genital organs (smiling). Many of my friends are afraid to take the vaccine due to this fact. We need to get married and have kids in the future. Who can guarantee that there are no such effects of the vaccines…(smiling)*”(female, age 22, urban, tertiary education, IT officer in profession)

As revealed, this fear of adverse impacts is encountered due to various instances of misinformation and disinformation shared by various parties, including health sector professionals, peers and friends, community members, and family members. There is a considerable gap between the communities and their age, ethnicity, and living status regarding the prevalence of knowledge of potential side effects of vaccines rather than presumed side effects.

### 3.3. Public Health Information and Related Factors Affecting Vaccine Intake by the Respondents

The role of public media and other information sources is significant in disseminating public health information during any health emergency. Only 25.09% (*n* = 3113) of respondents in the sample believed that they make decisions regarding vaccines based on the scientific information received from trusted channels. In comparison, around 38% of respondents have no concerns about vaccine information when making vaccine decisions. In addition, around 37% of respondents do not know the importance of seeking scientific information. Television and radio programmes conducted by health professionals were the most accepted and trusted source by respondents (25.09%) seeking scientific information regarding vaccines and vaccination, followed by social media sources (Appendix A).

When comparing seeking scientific knowledge against gender and economic variables, females were less likely not to make decisions based on scientific information compared to males (AOR 0.65) (Table 5), and middle-income groups were less likely not to make decisions based on scientific information and have no idea compared to underprivileged people (AOR 0.42 and AOR—0.62) (Table 6). This gap is encountered due to extreme inequality of information infrastructures among distinct economic settings.

According to the respondents, social media platforms have provided the most debatable and untrusted information from different groups, including professionals from local and international contexts. In this milieu, the survey explored the means of cross-checking the information presented by various parties during the vaccination process. Only 27.91% (*n* = 3113) of respondents mentioned that they usually double-check the information posted on social media platforms, whereas 33.74% accepted that they do not cross-check, and 38.35% had no idea about the need for cross-checking information. Interestingly, interview data suggested that the majority of respondents admitted that they cross-check the information shared on social media, disclosed that ‘asking from a friend or peer’ is the most popular way of cross-checking, followed by ‘searching internet sources’ and ‘checking with YouTube channels published by Sri Lankans’. Further interview data revealed that individuals, especially young members, have provided inputs to family members based on content from known and unknown individuals on social media.

“*We have no idea about vaccines, the impact of vaccines, and presumed side effects, which people discuss in some informal forums. When we had something to confirm, we used to ask our younger son. He is searching Facebook and telling us what is right and wrong…”*(Female, age 54, rural, secondary education, housemate).

Accordingly, adults used their young children as a source of information, who have substantial knowledge and access to internet sources, mainly social media, to gather information and double-check the accuracy of burning matters.

In terms of the sector of residents, people from the estates were less likely to make vaccine decisions based on scientific information compared to village people (AOR—2.97) (Table 7). In contrast, urban sector respondents were more concerned about decision-making based on scientific information since their likelihood of making decisions without scientific information was low compared to village people (AOR 0.79). This suggests that people in urban settings tend to make decisions based on scientific information compared to other sectors. Interview data indicated that people from rural and estate sectors, especially underprivileged groups in such settings, have suffered from an extreme lack of information facilities, such as networks, devices, and adequate data bundles.

#### Factors Affecting Satisfaction with Public Health Information Sharing among Respondents Regarding Vaccines

Based on several criteria, the study investigated peoples’ opinions about the public health information received during the pandemic regarding satisfaction, public trust, and causes. Results suggest that 70% (*n* = 3113) of respondents are not adequately satisfied with the information they received from authorities on vaccines and the impact of vaccination. The majority (48.58%) were in a moderate position regarding the level of satisfaction with the information received regarding the vaccination process. Only around 27% of respondents claimed they received satisfactory knowledge, which facilitated them to make necessary decisions during the COVID-19 pandemic.

Several key reasons that affect the level of satisfaction with the public health information received during the pandemic were investigated, and respondents’ opinions are highlighted in Figure 4. 

Respondents were allowed to record multiple reasons according to their views to record their answer. Overall, from the survey sample (*n* = 3113), the inadequacy of the public health information received on vaccines and vaccination from the health authorities during the pandemic was the most cited reason (57.80%) to be dissatisfied. The second most cited reason for dissatisfaction with the information was difficulty understanding the content of the message due to technical/medical terms in the health messages disseminated by the authorities, emphasised by 53.32% of respondents who mentioned that they are unsatisfied with the information they received. A significant percentage of respondents (42.63%) disclosed that the delay in receiving health information was a critical factor that impacted their decision-making. Further, 31.9% of respondents from the same cohort believed that a lack of trust in authorities significantly impacts vaccine hesitancy (Appendix A). Interview data highlighted that the inconsistency of health messages delivered by different health authorities was a significant factor in navigating distrust of health messages.

“*I am involved in public health-related activities at the local level. We are a group that works closely with the community. People ask all doubtful matters from us as they trust us. However, we also faced a challenging situation due to the inconsistency of messages delivered by authorities and officers. We know that COVID-19 is a new challenge, and many of us are learning things by doing. However, we also experienced pathetic contradictions since several officials publicly shared different views on some significant issues associated with COVID-19 in the first half of the pandemic…*”(Male, age 52, urban, higher education, public health inspector)

Many educated respondents believe that contradictory messages issued by national and regional health authorities in the same context contributed to reducing public trust during the COVID-19 pandemic. As the interviewees revealed, this contradiction was increased due to some health professionals sharing several private videos that contained anti-vaccine-related context.

## 4. Discussion

Numerous disease prevention and control strategies, including time-tested interventions like adhering to universal precautions, respiratory etiquette, and physical distancing, were practiced during the pandemic. However, given the adaptability and high transmissibility of the virus, a safer, more economical and potent solution was explored. Vaccination, an evidence-based intervention that has yielded results previously, was the preferred intervention by governments worldwide [47]. Subsequently, mass vaccination campaigns were initiated to cover the population at risk and finally achieve population-wide immunity or herd immunity [48]. Even though governments worldwide tried to vaccinate as many people as possible, recent multi-country research suggested that an average of 20.9% of people were reported to be hesitant to get the COVID-19 vaccine despite its wide availability [32]. This brings us to the question of why people refuse or are hesitant to accept the vaccine despite numerous studies proving it safe and efficacious [49,50]. Understanding vaccine hesitancy and its underlying factors would benefit policymakers by bridging the knowledge gap on vaccines and their processes among the public, thereby ensuring more public acceptance and trust based on solid evidence during future pandemics.

Adequate awareness and knowledge of the impact of the vaccines against the infection and its causative agent are significant factors in vaccine hesitancy. Many anti-vaccine campaigns have been launched based on false or presumed negative impacts of vaccines [14]. Adverse events following immunisation (AEFI) was the most influential characteristic of vaccine acceptance in many countries during vaccination against COVID-19 [51]. The findings affirmed this factor, since the majority who have either delayed or hesitated to accept the vaccine mentioned that presumed side effects or adverse events after immunisation are the most worrying for them [52,53]. The fear of adverse events after immunisation has mainly been caused by various social media posts and rumours disseminated by different groups, which is inevitable during a global pandemic [13,22,54].

Surprisingly, even though the level of knowledge among respondents on the effectiveness of vaccines is considerably low, waiting to see the effectiveness of vaccines has been the second highest determinant of vaccine hesitancy informed by the respondents. This suggests that the effectiveness threshold was a key and highly influential factor of vaccine acceptance among many other countries [51].

Further, findings stressed that respondents have higher trust in government information and advice on vaccination in which professionals and public health institutions were involved. However, respondents’ satisfaction with health messages disseminated by health authorities had significant drawbacks due to discrepancies in content, technical jargon, and contradictions. Faith and trust in the public health care system, health authorities, and employers, identified as a positive determinant of vaccine acceptance, is an advantage to building trust among people towards vaccination [32,55]. For example, vaccine acceptance is very high among some Asian countries such as China, South Korea, and Singapore, where people have strong trust in the government and local health systems [32]. On the contrary, some countries experienced jeopardization in the vaccination process due to politicisation, which resulted in inequality of vaccination and unrest among people [56].

Another aspect of hesitancy observed during the survey was hesitancy toward specific vaccines. Therefore, one group of respondents had spent a considerable amount of time accepting vaccines until they received their preferred vaccine brand, seen mostly among educated and young respondents. Interview data suggest that the attitudes have caused this preference and these perceptions of respondents, who believe that some vaccine manufacturers and countries are better at producing vaccines. For example, a set of questions was posed to the respondents to identify vaccine preferences. Results show that many have chosen one vaccine as their first, second, and third preferences (Appendix A). Interestingly, the preferable vaccine was not the highest administrated vaccine in the country, and it was not even the most-known vaccine among the respondents. Most respondents were unaware of vaccine manufacturing technologies, manufacturers, and preferences. This was seen because of a lack of public health information sharing at local levels. However, misinformation and disinformation, largely encountered on social media, misled people due to this information gap [16]. Vaccine characteristics, such as manufacturing country, duration of protection, doses required, and technology used have been identified as vital factors influencing vaccine acceptance [51].

At the onset of the pandemic, there was a substantial demand for any type of vaccine, to get the initial protection against the virus. However, when the pandemic was progressing and with the availability of different types of vaccines in the market, people tended to demand a particular type of vaccine of their choosing (for example, mRNA vaccines as opposed to killed vaccines). However, the government’s intention was to vaccinate as many populations as possible to achieve herd immunity and bring a particular killed vaccine into substantial communities. This created a dip in the demand for this particular vaccine, hence thousands of doses bought for the third and fourth doses of the vaccine had to be discarded due to non-utilisation [39,42].

Understanding the role of opinions of primary networks, such as peers and friends, neighbourhood, extended families, and personal professional networks, is significant, as the influence of primary interactions determines vaccine acceptance [57]. As discussed above, ethnicity and religion in Sri Lanka are two significant determinants of vaccine acceptance [41]. Our findings highlighted that religious and cultural belief plays a significant role in vaccine acceptance in local contexts. For example, one religious community has refrained from accepting vaccines in many study areas due to their religious sentiments. During the COVID-19 pandemic, Sri Lanka struggled to impose a crematory policy for COVID-19 deaths, which was discriminatory for one religious community as they religiously restrict cremation. This has been heatedly debated in the national and international policy forums as ethnic discrimination and a fundamental right violation [46]. However, according to findings, particular religious communities broadly accepted vaccines after the country imposed the compulsory cremation, and again, vaccine acceptance dramatically decreased once the government declared that COVID-19 bodies could be buried alternatively. This pattern shows the significance of understanding neglected cultural factors, such as religious faith, that affect vaccine hesitancy. Therefore, the country needs a strong approach to promoting public partnership with policymaking towards effective vaccination [37,56,58].

Overall, the findings suggest that fear and lack of knowledge of the potential side effects of the vaccines had a significant relationship with vaccine hesitancy. Sri Lankan public health authorities have conducted studies on the adverse impacts of the vaccines and shared their results publicly [53]. Nevertheless, the results of this study revealed that the use of this scientific knowledge in vaccine acceptance and public access to this evidence-based knowledge was extremely low due to the lack of information facilities and other cultural factors, i.e., literacy and technicality of information. The idea of possible side effects, over exacerbated to create doubts, seemed cultivated by misinformation and disinformation circulated by local and international anti-vaxxers and other influencers [13,54]. Many of the above factors are linked to the lack of public knowledge about vaccines and vaccination and inadequate information shared by public health authorities [59]. Public attitudes towards the positive impacts of vaccines have been severely manipulated by misinformation and disinformation [13,21].

Many of determinants of the vaccine hesitancy discussed above have a strong connection with an infodemic [11]. Too much information, including false or misleading information in digital and physical environments during a disease outbreak, is defined as an ‘infodemic’ [9], a critical and common agent of vaccine hesitancy among many countries [60]. As per the study findings, dissemination of fake news and rumours through social media, particularly Facebook, TikTok, and WhatsApp, has a detrimental impact on vaccine acceptance, especially among young cohorts and their dependents [22,61]. Thus digital communication had a tremendous role in information management in the COVID-19 era [62]. In the Sri Lankan context, disinformation was widely shared by mainstream media sources, i.e., television and radio channels, and social media, which people largely accessed to obtain information, without critical assessments of the content released [63].

Like most other countries, the government has been the main source of health information in Sri Lanka [62]. Information sharing was designated to two institutions: the Government Information Department under the Ministry of Mass Media and the Health Promotion Bureau (HPB) under the Ministry of Health. The Government Information Department was tasked with giving daily updates on the number of patients, deaths, and any special announcements from either the Governement of Sri Lanka (GoSL) or National Operations Centre for Prevention of COVID-19 Outbreak (NOC PCO), while the HPB was assigned to risk communication and specific health-related information sharing. Both maintained a social media presence and the HPB had tailormade social media portals through Facebook, WhatsApp, and Viber to communicate health messages to the masses. Government Information Department media messages on COVID-19 patient numbers and deaths were considered official and other media outlets could disseminate the news after the Government Information Department published their daily reports. However, the government still needs to identify necessary gaps in the existing law, in terms of establishing new legal provisions to act effectively against people and agencies disseminating misinformation and disinformation during a health emergency [63].

Further, study revealed that inadequate information related to vaccination, which influences the decision to accept vaccines, was primarily seen among communities in different forms. Based on the findings of the study, we understand that there are two visible forms of information contexts that affect people’s decisions on vaccination.

Low density of information: estate communities, Indigenous communities, and some communities in remote rural areas and low-income settings have comparatively limited access to information sources due to a lack of information infrastructures, such as internet facilities, devices, and accessibility to reliable data, and lack of public health professionals to engage in such information sharing and discussions. Further, people with insufficient education levels to understand health information shared by different sources also suffer from a scarcity of information on the cause. Hence, rumours are highly prevalent in such settings due to inadequate information, which causes a dilemma of “*what to believe and what not*”.High density of information: Youth, educated people, and some communities, such as urban, semi-urban, and rural areas near urban centres, have too much information coming from various sources, such as internet-based channels, electronic media, social media, professional networks, and primary community networks. In this milieu, people have too much information, complicating their decision-making.

Policy priorities for international and national health authorities to respond to anti-vaccine lobbies are pivotal in improving vaccine decisions among the public [14,22]. However, results highlighted an unsatisfactory public response to ‘counter-anti-vaccine’ platforms and vaccine-related information-sharing mechanisms the government health authorities introduced. Knowledge sharing is challenging in resource-limited settings, such as remote rural areas, urban low-income settings, Indigenous communities, and estates. Therefore, these lessons are essential to make policy decisions on vaccination against future disease outbreaks in the country.

To address vaccine hesitancy and its complex maze of reasons, governments across the globe need to produce innovative and evidence-based interventions that can strengthen public trust and sustain the impetus of vaccine campaigns. These include (a) identifying and prioritise vaccine-hesitant population cohorts, (b) diagnosing the demand and supply side barriers and enabling factors of vaccination among these population cohorts, and (c) designing evident informed strategies to address vaccine hesitancy appropriate to context, culture, and other determinants of the population cohorts. By enabling such interventions to sustain the responses through community-oriented solutions, governments will be able to better respond to vaccine hesitancy and sustain high coverage to save lives.

Further studies need to understand the multidimensional uncertainty of vaccination in the Sri Lankan context, including possible scales, as our main intention of this paper is to discuss key social and economic determinants and the impact of public health information on vaccine hesitancy [64,65]. However, our study discusses certain aspects of multidimensional uncertainties, such as vaccine intention and gender [66], health risks, efficacy doubts, beliefs, lack of understanding, fear, and misinformation [65]. We believe that future studies on vaccine hesitancy behaviour in Sri Lanka can explore other critical dimensions, such as health risks, physical pain, inconvenience, personal reaction, and the influence of previous vaccines, i.e., influenza [64], which have not adequately been addressed in the current research literature.

## 5. Conclusions

In summary, fear of side effects of the vaccines was the main reason to reject or delay the vaccine followed by waiting to see the effectiveness of the vaccines and receiving negative information about the vaccines. The lack of information on the vaccine’s effectiveness against COVID-19, the outcome of the vaccines, and the adverse impacts of vaccines have a significant impact on the vaccine acceptance of respondents. Given the results elicited in this study, it is evident that the reasons for vaccine hesitancy are multifactorial. These factors include socio-economic, cultural, religious, and personal attributes. New phenomena like the spreading misinformation through media platforms by various groups with vested interests have further exacerbated the situation, adding to vaccine hesitancy. The strengths and weaknesses of a country’s health and public information system play a crucial role in such circumstances. A population with poor knowledge of the vaccine, vaccine development process, and possible adverse effects will further facilitate misinformation through gaps in the health and public information systems. All such factors combine to create a conducive environment for vaccine hesitancy.

Since the underlying causes of the problem are multifactorial, the solutions must be strong enough to handle numerous factors as well. Hence, rather than a blanket approach to addressing vaccine hesitancy, an all-inclusive, whole-of-society approach is warranted. Such a multi-sectoral approach addressing the political, social, economic, cultural, and personal attributes of vaccine hesitancy could be the way forward in infectious disease prevention in Sri Lanka.

We believe that this research is the first field-based study in Sri Lanka to investigate socio-economic and information determinants of the COVID-19 vaccine hesitancy, covering more than 3000 households. However, several limitations can influence the interpretation of the findings. As observed, many respondents who had underlying health problems or were on medication for chronic diseases seemed reluctant to accept vaccines or provide reasons for non-acceptance, hence they are underrepresented in the study. Another limitation of this study is an underrepresentation of respondents who are immunocompromised, due to ethical reasons involved in data collection considering various underlying health conditions. Further, income, age, and vaccination status revealed by the respondents were solemnly self-claimed and not cross-checked by other sources. The findings of the research on the implications of health information dissemination may not be directly applicable to countries that had effective communication with the public during vaccination. Also, determinants of hesitancy, such as religious beliefs and trust in traditional medicine and remedies, may be less relevant to some other countries as they are unique to the context. However, further research can address such specific determinants of their own settings as determinants of vaccine hesitancy are not common to every social setting and contextualisation is a must to understand the underlying factors that drive vaccine hesitancy.

## Figures and Tables

**Figure 1 ijerph-21-01268-f001:**
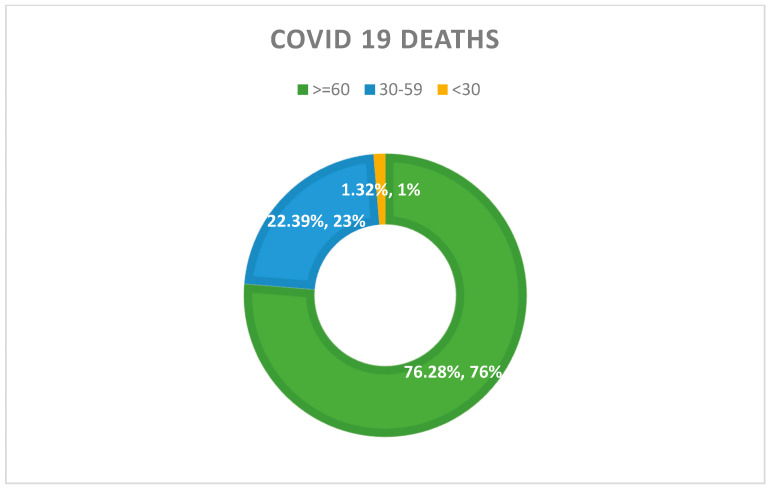
COVID-19 deaths in Sri Lanka by age cohorts. Source: The figure was developed based on the data presented by the Ministry of Health, Sri Lanka [35].

**Figure 2 ijerph-21-01268-f002:**
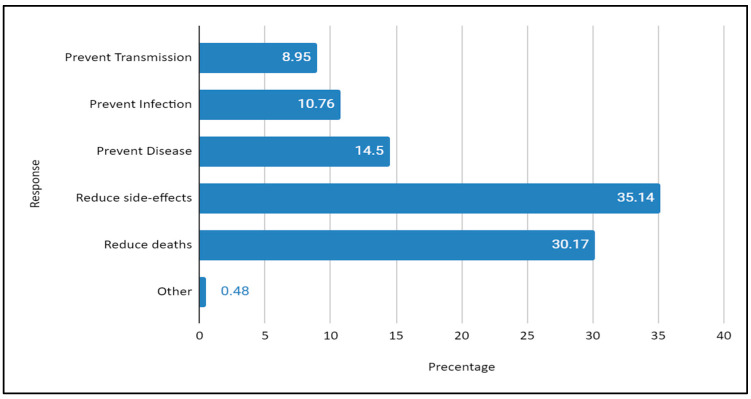
Responses on knowledge of the functions of vaccines against COVID-19. Source: field data.

**Figure 3 ijerph-21-01268-f003:**
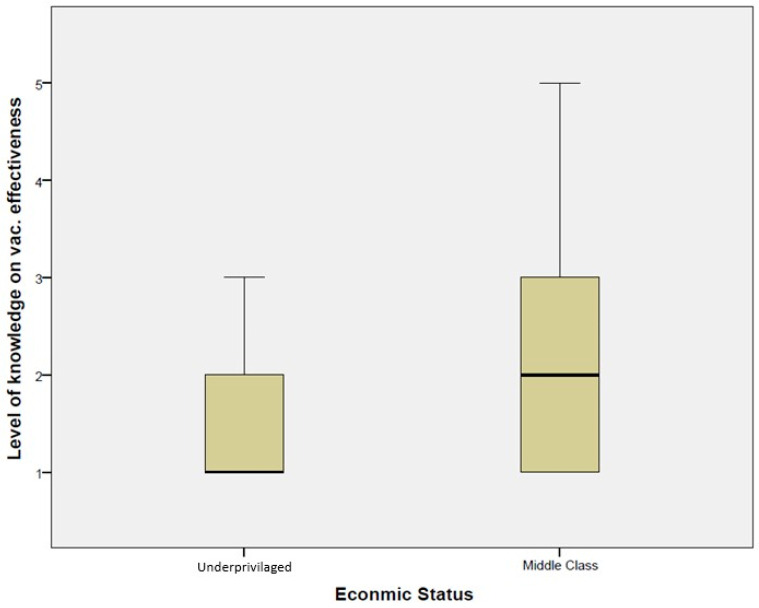
Difference of knowledge level on the effectiveness of the vaccines between the underprivileged and middle-income communities. Source: field data.

**Figure 4 ijerph-21-01268-f004:**
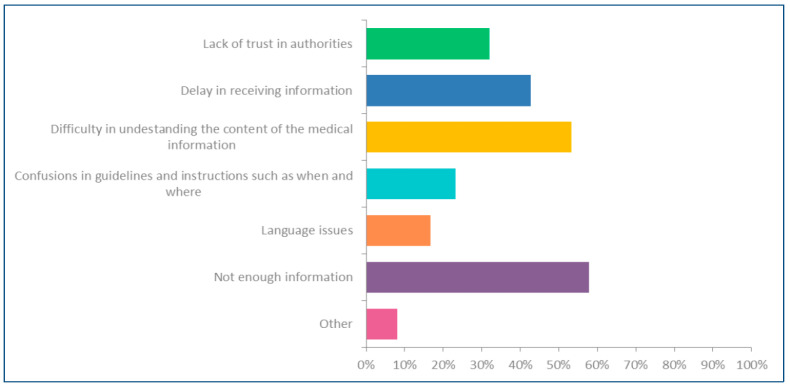
Reasons for being unsatisfied about the health information received during the COVID-19 pandemic. Source: filed data.

**Table 1 ijerph-21-01268-t001:** Number of doses administrated by vaccine types in Sri Lanka.

Vaccine	Dose	Number
AstraZeneca	1st dose	1,479,631
	2nd dose	1,418,593
Sinopharm	1st dose	12,054,824
	2nd dose	11,267,138
Sputnik V	1st dose	159,110
	2nd dose	155,812
Pfizer	1st dose	2,645,395
	2nd dose	1,123,923
	1st booster	8,220,002
	2nd booster	202,571
Moderna	1st dose	804,801
	2nd dose	787,361

Source: [35].

**Table 2 ijerph-21-01268-t002:** Districts and divisional secretariat (DS) areas selected for data collection.

District	DS Divisions
Kandy	Delthota, Pathadumbara, Yatinuwara, Nawalapitiya
Galle	Baddegama, Karandeniya, Kosgoda, Ahungalle
Matara	Mulatiyana
Anuradhapura	Mihinthale, Anuradhapura
Puttalam	Dankotuwa, Wennappuwa, Naththandiya
Kurunegala	Ibbagamuwa, Mallwapitiya, Kurunegala
Colombo	Padukka, Seethawaka, Thibirigasyaya, Athulkotte, Mahawatta
Kalutara	Dodangoda, Walaliawita
Badulla	Dambana, Rideemaliyadda

**Table 3 ijerph-21-01268-t003:** Vaccine shots accepted by respondents.

Vaccine Dose	Responses
Only first dose	7.53%
Only up to the second dose	55.25%
Only up to the first booster	32.47%
None	4.75%

Source: field data.

**Table 4 ijerph-21-01268-t004:** Reasons for delaying or rejection of vaccines.

	Reasons Admitted by Respondents	*n*
1	Lack of knowledge of vaccination	4.27%
2	Fear of presumed side effects	46.29%
3	Believing in traditional medicine or religious faith	3.95%
4	Waiting to see the results of vaccines (wait and see)	17.40%
5	Advised by someone not to take the vaccine	4.44%
6	Receiving negative information about the vaccine	12.46%
7	No vaccine centres nearby	3.69%
8	No vaccines available	3.06%
9	Lack of encouragement from authorities	2.37%
10	Other	1.67%

Source: field data.

**Table 5 ijerph-21-01268-t005:** Relationship between sex and knowledge of vaccines and use of scientific information.

Variables		Male (Reference)	Female	COR (95% CI)		*p*	AOR (95% CI)		*p*
Vaccine effectiveness	No knowledge (Ref)	725 (53.19)	927 (52.97)						
	Very low	308 (22.60)	391 (22.34)	0.99	(0.81–1.17)	0.94	1.05	(0.87–1.24)	0.58
	Low	253 (18.56)	294 (16.80)	0.91	(0.71–1.10)	0.33	0.92	(0.71–1.13)	0.43
	Moderate	69 (5.06)	126 (7.20)	1.43	(1.12–1.74)	0.02	1.31	(0.96–1.65)	0.13
	High	8 (0.59)	12 (0.69)	1.17	(0.27–2.07)	0.73	1.29	(0.38–2.20)	0.59
Side effects of the vaccine	No knowledge (Ref)	527 (38.66)	756 (43.20)						
	Very low	404 (29.64)	457 (26.11)	0.79	(0.61–0.96)	0.01	0.84	(0.66–1.02)	0.06
	Low	269 (19.74)	265 (15.14)	0.69	(0.48–0.89)	<0.01	0.74	(0.53–0.95)	0.01
	Moderate	127 (9.32)	194 (11.09)	1.06	(0.81–1.31)	0.62	1.00	(0.71–1.28)	0.98
	High	36 (2.64)	78 (4.46)	1.51	(1.10–1.92)	0.05	1.23	(0.78–1.68)	0.37
Vaccine suitability against COVID-19	No knowledge (Ref)	895 (65.66)	1101 (62.91)						
	Very low	254 (18.64)	311 (17.77)	1.00	(0.81–1.18)	0.96	1.02	(0.83–1.22)	0.83
	Low	154 (11.30)	205 (11.71)	1.08	(0.86–1.31)	0.50	1.04	(0.79–1.28)	0.77
	Moderate	50 (3.67)	115 (6.57)	1.87	(1.53–2.21)	<0.01	1.81	(1.43–2.18)	<0.01
	High	10 (0.73)	18 (1.03)	1.46	(0.69–2.24)	0.34	1.50	(0.71–2.29)	0.32
Decisions based on scientific information	Yes	353 (25.90)	561 (32.06)						
	No	553 (40.57)	561 (32.06)	0.64	(0.46–0.82)	<0.01	0.654	(0.47–0.84)	<0.01
	No idea	457 (33.53)	628 (35.89)	0.87	(0.69–1.04)	0.11	0.829	(0.63–1.03)	0.065

AOR—adjusted for income level, living area, age, religion. Source: field data

**Table 6 ijerph-21-01268-t006:** Relationship between income level and knowledge of vaccines and use of scientific information.

Variables		Underprivileged (Reference)	Middle Income	COR (95% CI)		*p*	AOR (95% CI)		*p*
Vaccine effectiveness	No knowledge (Ref)	1004 (64.73)	648 (41.49)						
	Very low	308 (19.86)	391 (25.03)	1.97	(1.79–2.15)	<0.01	1.58	(1.39–1.78)	<0.01
	Low	206 (13.28)	341 (21.83)	2.56	(2.37–2.76)	<0.01	2.12	(1.90–2.34)	<0.01
	Moderate	31 (2.00)	164 (10.50)	8.20	(7.80–8.59)	<0.01	8.16	(7.72–8.59)	<0.01
	High	2 (0.13)	18 (1.15)	13.94	(12.48–15.41)	<0.01	9.85	(8.37–11.33)	<0.01
Side effects of the vaccine	No knowledge (Ref)	795 (51.26)	488 (31.24)						
	Very low	400 (25.79)	461 (29.51)	1.88	(1.70–2.05)	<0.01	1.33	(1.14–1.52)	<0.01
	Low	225 (14.51)	309 (19.78)	2.24	(2.03–2.44)	<0.01	1.49	(1.27–1.71)	<0.01
	Moderate	77 (4.96)	244 (15.62)	5.16	(4.88–5.44)	<0.01	4.12	(3.80–4.44)	<0.01
	High	54 (3.48)	60 (3.84)	1.81	(1.43–2.19)	<0.01	1.33	(0.91–1.76)	0.19
Vaccine suitability against COVID-19	No knowledge (Ref)	1154 (74.40)	842 (53.91)						
	Very low	234 (15.09)	331 (21.19)	1.94	(1.75–2.13)	<0.01	1.85	(1.64–2.06)	<0.01
	Low	128 (8.25)	231 (14.79)	2.47	(2.24–2.71)	<0.01	2.33	(2.07–2.59)	<0.01
	Moderate	32 (2.06)	133 (8.51)	5.70	(5.30–6.09)	<0.01	6.06	(5.63–6.50)	<0.01
	High	3 (0.19)	25 (1.60)	11.42	(10.22–12.62)	<0.01	16.37	(15.00–17.73)	<0.01
Decisions based on scientific information	Yes	313 (20.18)	601 (38.48)						
	No	625 (40.30)	489 (31.31)	0.41	(0.23–0.59)	<0.01	0.42	(0.23–0.62)	<0.01
	No idea	613 (39.52)	472 (30.22)	0.40	(0.22–0.58)	<0.01	0.64	(0.43–0.85)	<0.01

AOR—adjusted for gender, living area, age, religion; source: field data.

**Table 7 ijerph-21-01268-t007:** Relationship between living area and knowledge of vaccines and use of scientific information.

Variables		Rural (Ref)	Urban	Estate	Urban			Estate			Urban			Estate		
					COR (95% CI)		*p*	COR (95% CI)		*p*	AOR (95% COI)			*p*	AOR (95% COI)	*p*.
Vaccine effectiveness	No knowledge (Ref)	1043 (50.46)	233 (40.66)	376 (79.49)												
	Very low	489 (23.66)	143 (24.96)	67 (14.16)	1.31	(1.07–1.54)	0.02	0.38	(0.10–0.66)	<0.01	1.29	(1.04–1.54)	0.04	1.01	(0.66–1.37)	0.94
	Low	379 (18.34)	147 (25.65)	21 (4.44)	1.74	(1.50–1.97)	<0.01	0.15	(−0.30–0.61)	<0.01	1.54	(1.27–1.80)	<0.01	0.48	(−0.04–1.01)	0.01
	Moderate	140 (6.77)	46 (8.03)	9 (1.90)	1.47	(1.11–1.83)	0.04	0.18	(−0.51–0.86)	<0.01	1.45	(1.03–1.86)	0.08	1.69	(0.90–2.49)	0.19
	High	16 (0.77)	4 (0.70)	0 (0.0)	1.12	(0.01–2.22)	0.84	0.00	(0.00–0.00)	<0.01	1.41	(0.28–2.54)	0.55	0.00	(0.00–0.00)	<0.05
Side effects of the vaccine	No knowledge (Ref)	769 (37.20)	155 (27.05)	359 (75.90)												
	Very low	599 (28.98)	192 (33.51)	70 (14.80)	1.59	(1.35–1.83)	<0.01	0.25	(−0.03–0.53)	<0.01	1.53	(1.28–1.78)	<0.01	0.53	(0.18–0.89)	<0.01
	Low	394 (19.06)	111 (19.37)	29 (6.13)	1.40	(1.13–1.67)	0.02	0.16	(−0.24–0.56)	<0.01	1.38	(1.09–1.67)	0.03	0.38	(−0.09–0.85)	<0.01
	Moderate	219 (10.60)	89 (15.53)	13 (2.75)	2.02	(1.72–2.32)	<0.01	0.13	(−0.45–0.70)	<0.01	1.92	(1.57–2.26)	<0.01	0.87	(0.21–1.53)	0.68
	High	86 (4.16)	26 (4.54)	2 (0.42)	1.50	(1.03–1.97)	0.09	0.05	(−1.36–1.46)	<0.01	1.21	(0.68–1.74)	0.48	0.24	(−1.34–1.81)	0.07
Vaccine suitability against COVID-19	No knowledge (Ref)	1300 (62.89)	334 (58.29)	362 (76.53)												
	Very low	383 (18.53)	101 (17.63)	81 (17.12)	1.03	(0.78–1.28)	0.84	0.76	(0.49–1.03)	0.04	1.10	(0.83–1.36)	0.49	2.68	(2.32–3.03)	<0.01
	Low	248 (12.00)	91 (15.88)	20 (4.23)	1.43	(1.16–1.70)	0.01	0.29	(−0.18–0.76)	<0.01	1.46	(1.16–1.76)	0.01	1.61	(1.04–2.18)	0.10
	Moderate	117 (5.66)	41 (7.16)	7 (1.48)	1.36	(0.99–1.74)	0.11	0.21	(−0.56–0.99)	<0.01	1.48	(1.05–1.91)	0.07	3.09	(2.24–3.94)	0.01
	High	19 (0.92)	6 (1.05)	3 (0.63)	1.23	(0.30–2.15)	0.66	0.57	(−0.66–1.79)	0.36	1.49	(0.54–2.45)	0.41	3.97	(2.42–5.53)	0.08
Decisions based on scientific information	Yes	663 (32.08)	203 (35.43)	48 (10.15)												
	No	767 (37.11)	232 (40.49)	115 (24.31)	0.99	(0.77–1.20)	0.91	2.07	(1.72–2.42)	<0.01	1.06	(0.82–1.30)	0.63	0.79	(0.33–1.25)	0.31
	No idea	637 (30.82)	138 (24.08)	310 (65.54)	0.71	(0.47–0.95)	0.01	6.72	(6.40–7.05)	<0.01	0.79	(0.52–1.06)	0.09	2.97	(2.55–3.39)	<0.01

AOR—adjusted for income level, gender, age, religion; source: field data.

## Data Availability

The original contributions presented in the study are included in the article/Appendix A; further inquiries can be directed to the corresponding author.

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
