# Peer review of "Panic in the Pandemic: Determinants of Vaccine Hesitancy and the Dilemma of Public Health Information Sharing during the COVID-19 Pandemic in Sri Lanka"

_ijerph, 2024, doi:10.3390/ijerph21101268_

Round 1

Reviewer 1 Report

Comments and Suggestions for Authors

The paper deals with vaccine hesitancy, a pressing global challenge

The study aims to explore the socio-economic determinants that have influenced vaccine hesitancy and   the impact of public health information sharing in Sri Lanka during the pandemic. The study employs a comprehensive mixed-method approach for data collection, administrating household survey (n=3330) and 206 semi-structured interviews Survey results indicated that 37.8% (n=3113) of 20 respondents delayed or rejected vaccines for various reasons, the leading cause being the fear of 21 side effects of the vaccine. Although fear of side effects was the prime reason for rejection (n=1176, 22 46.29%), respondents demonstrated an extremely poor understanding of the potential side effects of vaccines, which was 55.39% (n=3113). Notably, 84.60% (n=3113) were unaware of the vaccine development process. The survey also revealed that strong belief in the use of traditional medicines and remedies and religious beliefs (n=1176, 3.95%) were among the main reasons for hesitancy. The findings illustrate that misinformation, lack of health knowledge, and lack of understanding to seek scientific information have fuelled vaccine hesitancy in Sri Lanka.

First let me compliment the author for the excellent paper.

The manuscript is well-grounded, offering a solid overview of vaccine hesitancy and is methodologically flawless.

Results are without any doubt interesting and this manuscript contributes to the global understanding of vaccine hesitancy in less studied countries.

Thus I recommend its publication.

Nonetheless,  I suggest MAJOR revision because I believe that the author(s) missed two seminal points in their overview. But I am sure that the authors will easily fix these flaws.

One of the most relevant findings is that “The proliferation of misinformation, including  false claims and rumours, and the influence of anti-vaccine movements have fuelled hesitancy”.

This issue is prevalent all over the world and deserves to be discussed in the overview.

There is already a wide literature on the topic, in particular referring to social networks (see for instance: https://doi.org/10.3390/socsci10080294 Rocha, Y. M., De Moura, G. A., Desidério, G. A., De Oliveira, C. H., Lourenço, F. D., & de Figueiredo Nicolete, L. D. (2021). The impact of fake news on social media and its influence on health during the COVID-19 pandemic: A systematic review. Journal of Public Health, 1-10)

Authors should summarize this literature (See) and provide the reader with a background on where the people of Sri Lanka got their info.

In the same vein, most studies reveal that Governments have been the main source of information for citizens worldwide. Accordingly, this topic should be introduced and discussed.

See:

https://doi.org/10.4185/rlcs-2023-1845

In particular, this reviewer missed some  specific info about how the Government of Sri Lanka dealt with the pandemic (info campaign, social media, etc.)

Good luck!

Author Response

Comments 1: [One of the most relevant findings is that “The proliferation of misinformation, including  false claims and rumours, and the influence of anti-vaccine movements have fuelled hesitancy”.

This issue is prevalent all over the world and deserves to be discussed in the overview.

There is already a wide literature on the topic, in particular referring to social networks (see for instance: https://doi.org/10.3390/socsci10080294 Rocha, Y. M., De Moura, G. A., Desidério, G. A., De Oliveira, C. H., Lourenço, F. D., & de Figueiredo Nicolete, L. D. (2021). The impact of fake news on social media and its influence on health during the COVID-19 pandemic: A systematic review. Journal of Public Health, 1-10)

Authors should summarize this literature (See) and provide the reader with a background on where the people of Sri Lanka got their info.]

Response 1: [We addressed this suggestion substantially in the discussion section in the revision (see newly added lines 732-756 and recommended references). Also, we included some additional readings in the introduction to enrich the aspect.]

Comments 2: [In the same vein, most studies reveal that Governments have been the main source of information for citizens worldwide. Accordingly, this topic should be introduced and discussed.

See:

https://doi.org/10.4185/rlcs-2023-1845

In particular, this reviewer missed some  specific info about how the Government of Sri Lanka dealt with the pandemic (info campaign, social media, etc.)]

Response 2: [Well addressed this recommendation also while revising the manuscript (see newly added lines 741-756)] 

Reviewer 2 Report

Comments and Suggestions for Authors

The article is very interesting and well-written. However, small revisions are needed:

1. The abstract is not focused on the main results. It would be beneficial to integrate information from the regression analysis, as this will support the study's aim of finding determinants.

2. The protocol number of ethical approval should be added.

3. The discussion section should include the limitations and strengths of the study.

4. The conclusion should support the main objective in the first sentence.

Author Response

Comments 1: [The abstract is not focused on the main results. It would be beneficial to integrate information from the regression analysis, as this will support the study's aim of finding determinants]

Response 1: [The abstract was revised accordingly and included significant information from regression analysis (see lines 25-27)]

Comments 2: [The protocol number of ethical approval should be added]

Response 2: [The protocol number has been added]

Comments 3: [The discussion section should include the limitations and strengths of the study]

Response 3: Well addressed this aspect in the conclusion section (see newly added section, lines 825-842)]

Comments 4: [The conclusion should support the main objective in the first sentence]

Response 4: [Substaintially addressed this by adding new lines 805-809]

Reviewer 3 Report

Comments and Suggestions for Authors

The paper is interesting and well written.

However:

The authors should describe the influence of multi-dimensional uncertainty on vaccine hesitancy, during the COVID-19 pandemic.

The authors should describe the influence of supply-side and demand-side factors on vaccine hesitancy.

In line 265 the term “discrete” should be replaced by the term “categorical”.

The lines 273-274 should be rewritten. The term “significance level” should be included.

In Table 4, the term “Responses” should be replaced by the letter “n”.

The authors should perform goodness of fit tests in Logistic Regression.

Author Response

Comments 1: [The authors should describe the influence of multi-dimensional uncertainty on vaccine hesitancy, during the COVID-19 pandemic]

Response 1: [addressed this aspect in the discussion, highlighting our contribution to multi-dimensional uncertainty (see newly included section, lines 793-803)]

Comments 2: [The authors should describe the influence of supply-side and demand-side factors on vaccine hesitancy.]

Response 2: [Sustaintially addressed this aspect by adding a new para (lines 686-694)]

Comments 3: [ In line 265 the term “discrete” should be replaced by the term “categorical”.]

Response 3: [replaced]

Comments 4: [The lines 273-274 should be rewritten. The term “significance level” should be included.]

Response 4: [Addressed]

Comments 5: [In Table 4, the term “Responses” should be replaced by the letter “n”.]

Response 5: [replaced]

Comments 6: [The authors should perform goodness of fit tests in Logistic Regression.]

Response 6: [Well addressed (see lines 420-428)]

Round 2

Reviewer 1 Report

Comments and Suggestions for Authors

The manuscript has improved and is now ready to be published